# Hysteresis Modeling and Compensation of Piezoelectric Actuators Using Gaussian Process with High-Dimensional Input

Yixuan Meng ![ID], Xiangyuan Wang, Linlin Li ![ID], Weiwei Huang and Limin Zhu *

State Key Laboratory of Mechanical System and Vibration, School of Mechanical Engineering, Shanghai Jiao Tong University, Shanghai 200240, China; mengyixuan@sjtu.edu.cn (Y.M.); wangxiangyuan@sjtu.edu.cn (X.W.); lilinlin321@sjtu.edu.cn (L.L.); 12510102161080@sjtu.edu.cn (W.H.)
* Correspondence: zhulm@sjtu.edu.cn

**Abstract:** Rate-dependent hysteresis seriously deteriorates the positioning accuracy of the piezoelectric actuators, especially when tracking high-frequency signals. As a widely-used nonparametric Bayesian method, the Gaussian process (GP) has proven its effectiveness in nonlinear hysteresis modeling. In this paper, the dimension of the input to the GP model is extended to consider more dynamic features of the tracking signal so as to improve the rate-dependent hysteresis modeling accuracy. In contrast with the traditional training set containing only the position and speed information, the acceleration and jerk information, as well as their temporal distribution information, is also included in the input of the model. An inverse hysteresis compensator (IHC) is established in the same way, and open-loop and closed-loop controllers are developed by using the IHC. Experimental results on a PEA stage show that with the increase in the input dimension, the hysteresis modeling accuracy improves greatly and, thus, the controllers based on IHC can achieve a better tracking performance.

**Keywords:** piezoelectric actuators; hysteresis modeling; Gaussian process; training input; tracking control

## 1. Introduction

Piezoelectric-driven nano-positioning stages are widely used in ultra-precision machining and measurement, such as fast tool servo (FTS) [1,2] and atomic force microscope (AFM) [3,4]. However, the rate-dependent hysteresis of the piezoelectric actuators (PEAs) seriously deteriorates the positioning accuracy, resulting in the morphological error of FTS and the image distortion of AFM [5]. In addition, as the input voltage frequency increases, the hysteresis loop becomes larger and rounder, which seriously limits the positioning performance, especially in high-speed positioning applications [6]. Therefore, in order to improve the positioning accuracy of the PEAs, it is important to model and compensate for the rate-dependent hysteresis nonlinearity.

In the past few decades, researchers have conducted extensive works to deal with the challenge of hysteretic nonlinearity modeling [7–9]. The relevant models can be generally divided into two types: the physics-based model and the phenomenological model [10,11]. The physics-based hysteresis model is derived from the basic physical principles of hysteresis materials via the relationship between physical quantities and empirical formulas [12]. Nevertheless, due to the complex physical causes of the actual nonlinear hysteresis system, it is difficult to establish the model based on physical principles. On the other hand, the phenomenological hysteresis model directly uses mathematical models to describe the nonlinear input and output relationship of hysteresis, without considering its inherent physical characteristics. Because of its relatively simple structure, the phenomenological model has become the most widely used one in hysteresis research, which mainly includes the Preisach model [13,14], Prandtl–Ishlinskii (P-I) model [15,16], Bouc–Wen model [17–19],

Dahl model [20], Duhem model [21], etc. Unfortunately, most of these models can only describe the rate-independent hysteresis.

Owning to the fact that hysteretic nonlinearity is generally rate-dependent [6], more rate-dependent models were developed to consider the changing rate of the input voltage, including the rate-dependent Preisach model [22] and rate-dependent PI model [23–26]. Nonetheless, these models usually contain plentiful unknown parameters, which makes the identification more difficult. Therefore, in 2015, Yang et al. [27] introduced a velocity damping mechanism into the traditional PI model, which successfully constructed a modified rate-dependent PI (MPI) model. This model can accurately describe the dynamic hysteresis characteristics with a relatively simple model structure and relatively few unknown parameters. Recently, with the development of machine learning methods, intelligent models are becoming more popular due to their powerful nonlinear approximation capability [28,29]. Researchers have developed various machine learning methods to model the rate-dependent hysteresis, such as neural networks [30], fuzzy system [31], least-squares support vector machines [32], adaptive fuzzy internal model [33] and so on. These intelligent methods have shown great improvements in modeling accuracy, which provide a new way for modeling the hysteresis of the PEAs.

As a widely-used machine learning method, the Gaussian process (GP), which is based on Bayesian probabilistic inference, removes the need for selecting many parameters while retaining the power of nonlinear dynamic description. Therefore, the usage of GP could make the model flexible and accurate without specifying the functional form and the parameters. It is a proper choice for rate-dependent hysteresis modeling. In 2019, Tao et al. [34] firstly applied the GP to the modeling and feedforward compensation of the PEA hysteresis. Since the hysteresis behavior of PEA is rate-dependent, the voltage value and its changing rate were utilized as a two-dimensional input of the training set and the displacement of the PEA was used as the output, which outperforms the aforementioned MPI model and many other classical models. However, for mixed-frequency signals that are widely used in nano-positioning tracking, the model performance is still unsatisfactory. Therefore, it is necessary to develop an effective model for complex tracking signals.

In this work, besides the position and velocity information used in the traditional models, the acceleration and jerk information, as well as their temporal distribution information, is introduced into the input of the training set and the dimension of the input is increased. The effect of input dimension on prediction is studied and the inverse hysteresis model is obtained via exchanging the input and output signals of the experiment. The IHC is established through the inverse hysteresis model, and open-loop and closed-loop controllers are developed by using the inverse hysteresis compensator (IHC). Comparative experimental investigations on a commercial piezo-actuated stage validate the effectiveness of the proposed GP-based model with high-dimensional input and the controllers based on the IHC. The results show that the increase in the input dimension would improve the accuracy of model prediction and the controller, especially for complex tracking signals. Therefore, the tracking performance of the controllers with the proposed model is greatly improved compared to those controllers with the traditional 2-dimensional GP model and the MPI model.

## 2. Rate-Dependent Hysteresis Nonlinear Modeling of PEA

For piezoelectric actuators, the mapping between the input voltage and the output displacement is nonlinear, rate-dependent and memorable because of hysteresis. Thus, the critical issue is to predict the output in the case of a particular input under the influence of hysteresis. The input and output of the GP training set can be expressed as $\mathbf{D} = \{X, y\} = \{x_i, y_i\}_{i=1}^{n}$, where $n$ is the length of the training input and output vectors and the training output $y_i \in \mathbb{R}$ is the measured displacement of the PEA. Different from the traditional two-dimensional input $x_i = (v_i, \dot{v}_i) \in \mathbb{R}^2$, where $v_i$ is the input voltage value and $\dot{v}_i$ is its relative changing rate, in this work, the dimensions of the input are extended to six, nine and twelve, which will be discussed in detail in Section 2.2. With these training sets,

the latent function $f(x)$ between the input and output could be mapped via training the GP model and the output displacement for a new input $X^*$ out of the training set can be predicted.

### 2.1. Principle of GP-Based Hysteresis Modeling

Essentially, GP is a series of random variables, in which any finite random variables are subject to joint Gaussian distribution over functions, $p(f)$. Therefore, if two or more points are picked in a function, observation of the outputs at these points will follow a joint Gaussian distribution. From a function-space view, a GP is formally specified by a mean function $m(x)$ and a covariance function $k(x, x')$, where:

$$m(x) = \mathbf{E}[f(x)] \tag{1}$$

$$k(x, x') = \mathbf{E}\big[(f(x) - m(x))(f(x') - m(x'))\big] \tag{2}$$

Therefore, the latent function of GP can be written as:

$$f(x) \sim \mathrm{GP}\big(m(x), \, k(x, x')\big) \tag{3}$$

In order to facilitate derivation, it is necessary to preprocess the data and subtract its mean to obtain the zero-mean distribution. The following inferences assume $m(x) = 0$. The covariance function $k(x, x')$, also called the kernel, plays a pivotal role in the Gaussian regression model.

Suppose that the data noise obtained by the sensor is $\varepsilon$, and assume it is independently and identically distributed and its variance is $\sigma_n^2$ [34], then:

$$y = f(x) + \varepsilon \tag{4}$$

and the joint prior distribution is:

$$\begin{bmatrix} y \\ y^* \end{bmatrix} \sim N\left( 0, \begin{bmatrix} K(X, X) + \sigma_n^2 I & K(X, X^*) \\ K(X^*, X) & K(X^*, X^*) \end{bmatrix} \right) \tag{5}$$

where $K(X, X)$ is the covariance matrix between all the data sets in input $X$ and it is expressed as:

$$K(X, X) = \begin{bmatrix} k(x_1, x_1) & \cdots & k(x_1, x_n) \\ \vdots & \ddots & \vdots \\ k(x_n, x_1) & \cdots & k(x_n, x_n) \end{bmatrix} \tag{6}$$

Other entries $K(X, X^*)$, $K(X^*, X)$ and $K(X^*, X^*)$ have similar definitions. According to the joint Gaussian prior distribution obtained from the observed value, the joint posterior distribution can be calculated as:

$$p(y^* | X, \, y, X^*) \sim N(\overline{y}^*, cov(y^*)) \tag{7}$$

It is the critical predictive equation for GP regression with:

$$\overline{y}^* \triangleq \mathbf{E}\big(y^* \big| X, \, y, X^*\big) = K(X^*, X)\Big[ K(X, X) + \sigma_n^2 I \Big]^{-1} y \tag{8}$$

$$cov(y^*) = K(X^*, X^*) - K(X^*, X)\Big[ K(X, X) + \sigma_n^2 I \Big]^{-1} K(X, X^*) \tag{9}$$

Finally, $\overline{y}^*$ will be taken as the prediction of the displacement of the PEA.

As mentioned above, the covariance function $k(x, x')$ is the kernel function of the GP, which defines the similarity between samples. As one of the most popular covari-

ance functions, the squared exponential (SE) function is chosen in this work, which is expressed as:

$$k(\boldsymbol{x}, \boldsymbol{x}') = \sigma_f^2 \exp\left(-\frac{||\boldsymbol{x} - \boldsymbol{x}'||^2}{2l^2}\right) \tag{10}$$

where $\sigma_f^2$ is the signal variance and $l$ is the length scale. These parameters contained in the covariance function are called hyperparameters, which are defined as $\boldsymbol{\theta}$. Considering the noise term $\varepsilon$ in Equation (4), the hyperparameters are:

$$\boldsymbol{\theta} = \left[l,\ \sigma_f^2,\ \sigma_n^2\right] \tag{11}$$

Hence, the logarithmic marginal likelihood is:

$$logp(\boldsymbol{y}|\boldsymbol{X},\ \boldsymbol{\theta}) = -\frac{1}{2}\boldsymbol{y}^T\left[K(\boldsymbol{X},\boldsymbol{X}) + \sigma_n^2\boldsymbol{I}\right]^{-1}\boldsymbol{y} - \frac{1}{2}\log\left|K(\boldsymbol{X},\boldsymbol{X}) + \sigma_n^2\boldsymbol{I}\right| - \frac{n}{2}log2\pi \tag{12}$$

The above logarithmic marginal likelihood can be maximized iteratively by the numerical optimization methods and the hyperparameters can be obtained using the training input information. For more information about the hyperparametric optimization, readers can refer to [34].

### 2.2. Extension of the Input Dimensions

As a rate-dependent behavior, the hysteresis nonlinearity is dependent on not only the voltage and its changing rate, but also the changing acceleration and even the jerk. Hence, in order to improve the modeling accuracy, it is necessary to increase the input dimension to include more information into the training set. Inspired by the work of Hu et al. to predict the contouring error dynamics of the multi-axis systems with a deep gated recurrent unit (GRU) neural network [35], the input dimension is extended to six, nine and twelve to improve the modeling performance.

Suppose that the voltage of PEA is:

$$\boldsymbol{v} = [v_T,\ v_{2T}, v_{3T}, \dots, v_{nT}]^T \tag{13}$$

where $T$ is the sampling period. By differentiating the voltage signal $\boldsymbol{v}$, the changing rate $\dot{\boldsymbol{v}}$, the changing acceleration $\ddot{\boldsymbol{v}}$ and the changing jerk $\dddot{\boldsymbol{v}}$ can be obtained as:

$$\dot{\boldsymbol{v}} = \left[\dot{v}_T,\ \dot{v}_{2T}, \dot{v}_{3T}, \dots, \dot{v}_{nT}\right]^T \tag{14}$$

$$\ddot{\boldsymbol{v}} = \left[\ddot{v}_T,\ \ddot{v}_{2T}, \ddot{v}_{3T}, \dots, \ddot{v}_{nT}\right]^T \tag{15}$$

$$\dddot{\boldsymbol{v}} = \left[\dddot{v}_T,\ \dddot{v}_{2T}, \dddot{v}_{3T}, \dots, \dddot{v}_{nT}\right]^T \tag{16}$$

Different from the two-dimensional input of the training set, which contains only the voltage and its changing rate information at time $t$, the six-dimensional input, nine-dimensional input and twelve-dimensional input at time $t$ are constructed by considering not only more temporal derivative information but also the temporal distribution information. They are described as follows:

$$\boldsymbol{x}_{6t} = \left(v_{t-T}, v_t, v_{t+T}, \dot{v}_{t-T}, \dot{v}_t, \dot{v}_{t+T}\right) \tag{17}$$

$$\boldsymbol{x}_{9t} = \left(v_{t-T}, v_t, v_{t+T}, \dot{v}_{t-T}, \dot{v}_t, \dot{v}_{t+T}, \ddot{v}_{t-T}, \ddot{v}_t, \ddot{v}_{t+T}\right) \tag{18}$$

$$\boldsymbol{x}_{12t} = \left(v_{t-T}, v_t, v_{t+T}, \dot{v}_{t-T}, \dot{v}_t, \dot{v}_{t+T}, \ddot{v}_{t-T}, \ddot{v}_t, \ddot{v}_{t+T}, \dddot{v}_{t-T}, \dddot{v}_t, \dddot{v}_{t+T}\right) \tag{19}$$

It is worth noting that the original signal collected by the displacement sensor includes noise. The differential process is that the displacement difference between two adjacent sampling points divides by the sampling period. If there exists displacement noise, the

noise will be enlarged greatly since it is divided by a rather small denominator, the sampling period. As the differential order increases, the noise will be consistently enlarged in the same way. Thus, a low-pass finite impulse response (FIR) filter is utilized to eliminate the noise of the training set.

### 2.3. Experimental Setup

The experiments are conducted on a commercial one-dimension piezoelectric-driven nano-positioning stage (P66X30, Harbin Core Tomorrow Science and Technology Co., Ltd., Harbin, China). Its displacement is realized by the deformation of a flexure hinge guiding mechanism with a travel range from 0~26.52 μm. The stage is driven by a piezo-ceramic-made PEA composed of lead zirconate titanate and an integrated high-resolution strain gauge sensor is used to measure the displacement. A voltage amplifier with an amplification factor of 15 is employed to drive the PEA and a signal conditioner is used to capture the position signal from the strain gauge sensor. The control algorithms are programmed in the environment of MATLAB/Simulink and downloaded to a rapid prototyping system (dSPACE-DS1103) equipped with several 16-bit Analog-to-Digital Converters (ADCs) and the 16-bit Digital to Analog Convertors (DACs) to realize real-time control. The sampling frequency of the experiments is selected as 20 kHz. The whole experimental setup is illustrated in Figure 1a, while its signal flow block diagram is shown in Figure 1b.

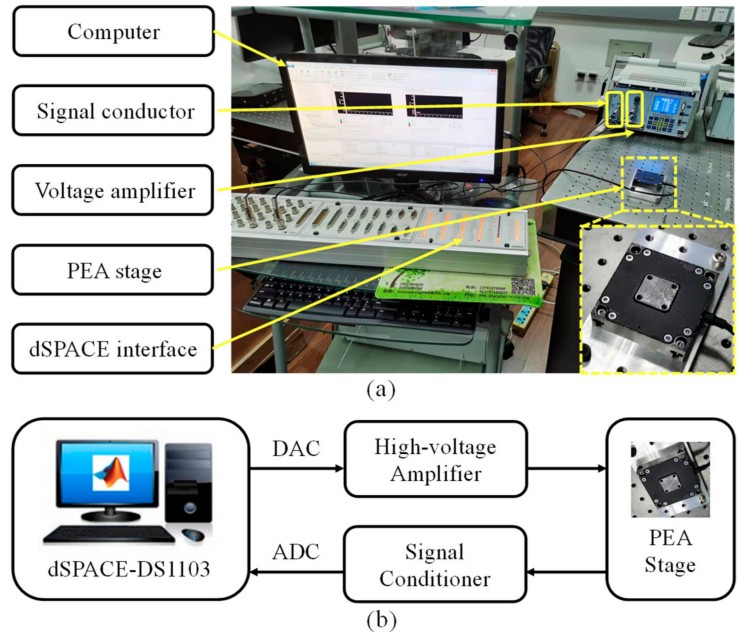

**Figure 1.** Experimental setup of PEA nano-positioning system: (**a**) experimental platform; (**b**) block diagram.

### 2.4. Modeling Results with Different Input Dimension

Since the bandwidth of the proportional-integral (PI) controller of the experimental stage is approximately 1700 Hz, 50–1000 Hz is selected as the interesting frequency range. Therefore, to obtain the training set, a chirp signal with a frequency from 25 to 1025 Hz and an amplitude of up to 30 V is selected as the voltage input to drive the PEA, as shown in Figure 2. And the displacement of the PEA stage is captured by the strain gauge sensor, which is used as the output of the training set. Additionally, the MPI rate-dependent model [27], a classical phenomenological model (please refer to the Appendix A for more details), is built to describe the hysteresis nonlinearity via the same dataset for comparison.

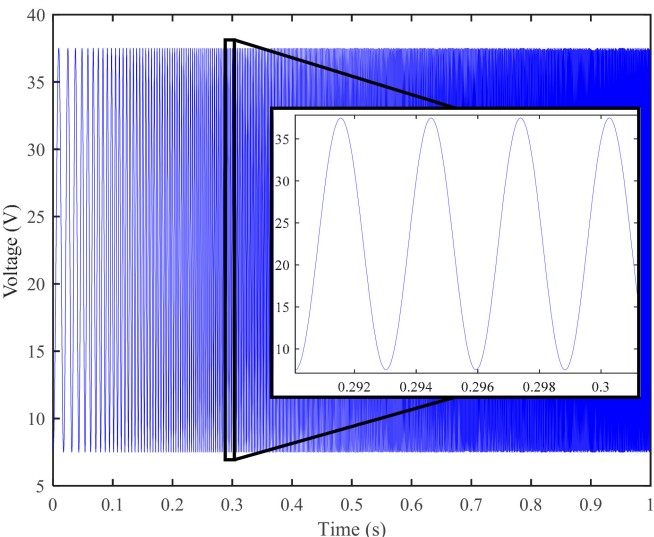

**Figure 2.** Driven signal for training.

To evaluate the modeling accuracy, the widely-used normalized root mean squared error (NRMSE) and the relative maximum error (RME) are chosen as the evaluation indexes [7,8,27,34], which are defined respectively as:

$$\text{NRMSE} = \frac{\sqrt{\frac{1}{n} \sum_{i=1}^{n} (y_i - \hat{y}_i)^2}}{\max_i(y_i) - \min_i(y_i)} \times 100\% \tag{20}$$

$$\text{RME} = \frac{\max_i(|y_i - \hat{y}_i|)}{\max_i(y_i) - \min_i(y_i)} \times 100\% \tag{21}$$

where $y_i$ and $\hat{y}_i$ are the true displacements and the model predictions of the PEA, respectively.

To evaluate the modeling accuracy with different input dimensions, sinusoidal signals with a frequency of 100 Hz, 200 Hz, 300 Hz, 500 Hz and 1000 Hz and an amplitude of up to 30 V, a two-frequency mixed-signal $x(t) = 7.5(\sin(120 \times 2\pi t) + \sin(180 \times 2\pi t)) + 22.5$, a four-frequency mixed-signal $x(t) = 6\sin(100 \times 2\pi t) + 3\sin(150 \times 2\pi t) + 3\sin(200 \times 2\pi t) + 6\sin(250 \times 2\pi t) + 22.5$ and a triangular wave of 50 Hz and an amplitude of up to 20 V, are chosen as the driven signals, respectively. It is noted that under high frequency inputs, the actual control voltage will be greatly enlarged due to the lightly-damped resonance of the nanopositioning stage, especially for the closed-loop controllers. To avoid the control saturation, this study is dedicated to the range from 0–10 μm as a reference for any range study. The hyperparameters are obtained by a typical GP algorithm implemented in our MATLAB environment, converged after some 200 iterations on the measured data. No specific parameter optimization is applied at this stage of the investigation that is focused on the adequate performance of the GP method. The results obtained with the MPI model and the GP-based model with different input dimensions are listed in Table 1, and a comparison results with different methods is shown in Figure 3.

From Table 1, it can be observed that for the sinusoidal signal inputs, the GP-based models with different input dimensions all outperform the MPI model. With the increase in the input dimension, the prediction accuracy is slightly enhanced. With the increase in the signal frequency, the predictive capability of the MPI model deteriorates rapidly. However, for the GP-based model, the prediction accuracy retains satisfactory even when the input frequency is very high. For the 1000 Hz sinusoidal signal, the NRMSE for the 12-dimensional input GP-based model is just 1.2855%, much smaller than that of the MPI model. For the mixed-frequency signals, the NRMSEs for the GP-based models with 2-dimensional and 6-dimensional inputs are similar to those for the MPI model. With the

increase in the input dimension, the prediction accuracy can be greatly improved. The NRMSE for the GP-based model with the 12-dimensional input is only 1.0759% for the four-frequency mixed-signal, which is just 37.26% of that for the MPI model and 35.62% of that for the GP-based model with the 2-dimensional input. For the triangular wave, the GP-based model yields the similar prediction accuracy as compared with the MPI model. The above testing results verify the effectiveness of GP-based rate-dependent hysteresis modeling with high-dimensional input and show that the modeling accuracy is greatly improved with the enhancement of the input dimension. The proposed high-dimensional GP-based model outperforms the classical phenomenological MPI model and the traditional 2-dimensional GP-based model for all testing signals. Compared with Refs. [7–9], this model method is capable of realizing more accurate prediction results with fewer hyperparameters. It has been proven a great success in modeling complex signals, including mixed-frequency signal and triangular wave. Nonetheless, the effectiveness of modeling these complex signals were not validated in these references.

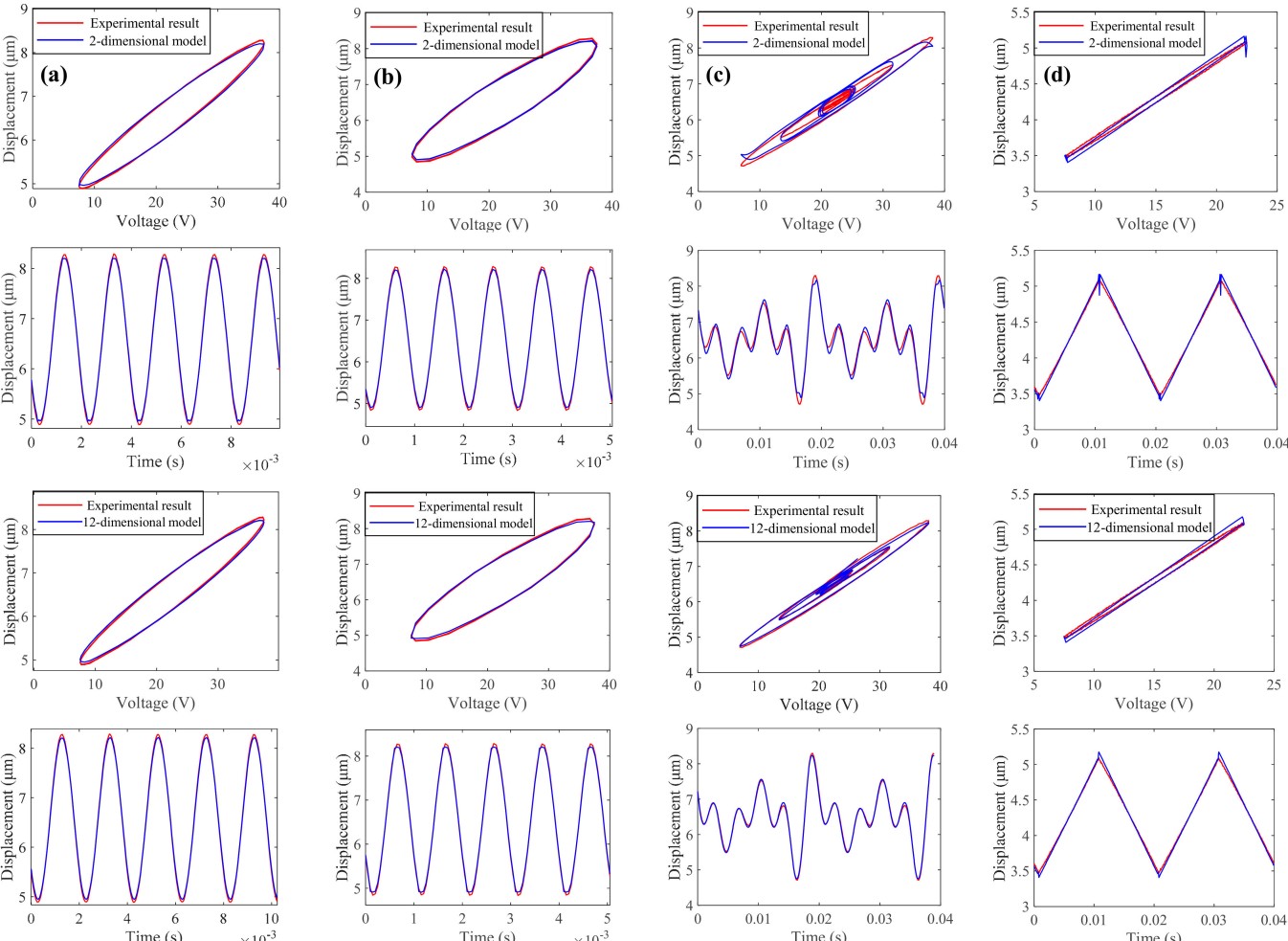

**Figure 3.** Experimental measured results of the PEA (red lines) as function of voltage and time for 500 Hz sinusoidal signal (column **a** panels); 1000 Hz sinusoidal signal (column **b** panels); four−frequency mixed−signal (column **c** panels); 50 Hz triangular wave (column **d** panels). The prediction results for 2−dimensional and 12−dimensional GP models are also plotted (blue lines) in the upper two and lower two panels, respectively.

**Table 1.** Experimental results for the MPI model and GP-based models with different dimensions.

| The Type of Input Signals | | MPI Model (NRMSE/RME, %) | GP-Based Model (NRMSE/RME, %) | | | |
| --- | --- | --- | --- | --- | --- | --- |
| | | | 2-Dimension | 6-Dimension | 9-Dimension | 12-Dimension |
| Sinusoid signal | 100 Hz | 0.6497/2.01 | 0.4034/0.95 | 0.1908/0.75 | 0.1812/0.66 | 0.1766/0.69 |
| | 200 Hz | 1.1673/2.93 | 0.2165/0.96 | 0.1712/0.56 | 0.1691/0.55 | 0.1673/0.56 |
| | 300 Hz | 2.9334/5.68 | 0.9519/1.93 | 0.9223/1.65 | 0.9187/1.65 | 0.9196/1.58 |
| | 500 Hz | 3.2692/6.33 | 1.3732/2.74 | 1.2974/2.31 | 1.3116/2.40 | 1.2749/2.29 |
| | 1000 Hz | 4.9547/8.96 | 1.3183/2.77 | 1.2875/2.32 | 1.2874/2.32 | 1.2855/2.31 |
| Mixed-frequency signal | 120 + 180 Hz | 2.1251/4.97 | 2.1670/4.65 | 2.1462/4.28 | 1.9929/3.83 | 1.6752/3.62 |
| | 100 + 150 + 200 + 250 Hz | 2.8872/6.69 | 3.0209/7.28 | 2.7079/6.19 | 1.6165/4.03 | 1.0759/2.76 |
| Triangular wave | 50 Hz | 2.1725/7.29 | 2.2333/13.8 | 2.1070/7.18 | 2.0444/7.41 | 2.0192/6.24 |

## 3. Controller Design Based on Hysteresis Compensation

As discussed above, the high-dimensional GP model can effectively predict the rate-dependent hysteresis. It can be used to correct the control of the PEAs via the IHC designed with the inverse GP model. Based on the IHC, open-loop and closed-loop controllers are constructed and tested to validate the effectiveness of the proposed model in this section.

### 3.1. The Open-Loop Controller Based on IHC

Owing to the fact that the inversion of hysteresis effect is by nature hysteresis loops, the direct inverse hysteresis compensation concept is widely used in the literature to help eliminate the hysteresis effect [36], which is shown in Figure 4. The compensation process can be written as:

$$y_{out}(t) = H\Big[H^{-1}[y_r]\Big](t) \tag{22}$$

where $H[\cdot]$ denotes the rate-dependent hysteresis model, $H^{-1}[\cdot]$ denotes its inverse model, $y_r(t)$ is the reference trajectory and $y_{out}(t)$ is the real output trajectory. The input voltage of the PEA, $v_{ff}(t)$, is generated from the IHC and the IHC can be directly modeled by GP with different input dimensions via interchanging the voltage and displacement as the input and output of the aforementioned GP model since the inverse of the hysteresis model is still a hysteresis model. The hyperparameters selected are the same as for Equation (11). For more details about the GP-based IHC modeling, readers can refer to [34].

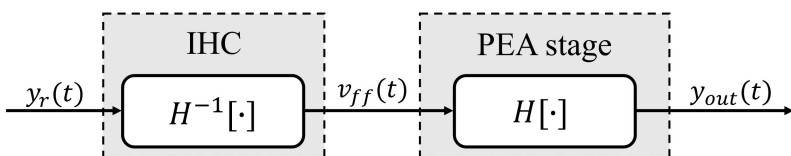

**Figure 4.** Block diagram of the open-loop controller based on the IHC.

To model the IHC so that $v_{ff}(t)$ can be obtained, the training data is chosen as $\tilde{\mathbf{D}} = \left\{\tilde{X}, v\right\} = \{\tilde{x}_i, v_i\}_{i=1}^n$, where $\tilde{x}_i$ is a two, six, nine or twelve-dimensional vector containing the PEA output displacement $y_i$ and its temporal derivative and temporal distribution information. For example, the 12-dimensional input $\tilde{x}$ at time $t$ is expressed as:

$$\tilde{x}_{12t} = \big(y_{t-T}, y_t, y_{t+T}, \dot{y}_{t-T}, \dot{y}_t, \dot{y}_{t+T}, \ddot{y}_{t-T}, \ddot{y}_t, \ddot{y}_{t+T}, \dddot{y}_{t-T}, \dddot{y}_t, \dddot{y}_{t+T}\big) \tag{23}$$

where $y$, $\dot{y}$, $\ddot{y}$ and $\dddot{y}$ are the displacement, velocity, acceleration and jerk of the PEA, respectively. For other dimensional input, $\tilde{x}$ can be constructed in a similar way.

For comparison, the inverse MPI model is built using the same training data as that used for the two-dimensional GP-based inverse hysteresis model. Sinusoidal sig-

nals with different frequencies from 100 Hz to 1000 Hz, a two-frequency mixed-signal $x(t) = 7.5(\sin(120 \times 2\pi t) + \sin(180 \times 2\pi t)) + 22.5$, a four-frequency mixed-signal $x(t) = 6\sin(100 \times 2\pi t) + 3\sin(150 \times 2\pi t) + 3\sin(200 \times 2\pi t) + 6\sin(250 \times 2\pi t) + 22.5$ and a triangular wave of 50 Hz are chosen as the reference trajectories, respectively. The testing results of the open-loop controllers with different compensators are listed in Table 2.

**Table 2.** Tracking results of the open-loop controllers.

| The Type of Reference Trajectories | | MPI-Based Compensator (NRMSE/RME, %) | GP-Based Compensator (NRMSE/RME, %) | | | |
|---|---|---|---|---|---|---|
| | | | 2-Dimension | 6-Dimension | 9-Dimension | 12-Dimension |
| Sinusoid signal | 100 Hz | 0.8204/1.80 | 0.5118/1.44 | 0.3565/1.01 | 0.5098/1.33 | 0.3178/0.96 |
| | 200 Hz | 1.5220/2.61 | 0.5622/1.50 | 0.5177/1.39 | 0.4199/1.25 | 0.5028/1.29 |
| | 300 Hz | 2.8829/4.67 | 1.7609/5.02 | 1.7623/3.60 | 1.3508/4.51 | 0.9646/2.05 |
| | 400 Hz | 3.1589/7.46 | 2.8196/8.65 | 1.9965/4.07 | 1.7018/3.84 | 1.0976/2.50 |
| | 500 Hz | 3.5382/12.1 | 2.7784/6.83 | 2.6411/5.74 | 2.0716/4.66 | 1.9783/4.85 |
| | 600 Hz | 5.4530/13.2 | 3.1666/7.58 | 2.8752/7.02 | 2.8545/6.14 | 1.4955/4.98 |
| | 700 Hz | 6.8551/19.0 | 3.5210/7.36 | 2.8407/8.30 | 2.5233/7.33 | 1.6366/5.48 |
| | 800 Hz | 7.9379/19.9 | 4.4801/10.0 | 3.5891/8.19 | 3.1453/7.77 | 2.7030/6.87 |
| | 900 Hz | 6.0733/16.1 | 5.5373/12.7 | 3.6697/8.93 | 3.7629/9.53 | 2.6164/6.67 |
| | 1000 Hz | 9.3307/19.6 | 4.6383/10.5 | 4.1660/9.26 | 4.0557/10.5 | 3.2564/8.85 |
| Mixed-frequency signal | 120 + 180 Hz | 1.6816/3.85 | 2.2292/7.72 | 1.2907/3.30 | 1.1217/2.83 | 1.1096/2.51 |
| | 100 + 150 + 200 + 250 Hz | 2.5270/8.65 | 3.5872/9.85 | 2.3128/5.79 | 1.9443/5.19 | 1.8159/4.86 |
| Triangular wave | 50 Hz | 2.0806/4.21 | 3.8415/9.04 | 1.0245/4.74 | 0.9673/2.78 | 0.8783/2.27 |

From Table 2, it can be observed that the GP-based compensators can provide better tracking accuracy for the sinusoidal signals as compared with the MPI compensator, especially when the tracking frequency becomes high. The 12-dimensional and 9-dimensional GP-based compensators outperform the 6-dimensional and 2-dimensional GP-based compensators. For the mixed-frequency signals and triangular wave, the 12-dimensional GP-based compensator and 9-dimensional GP-based compensator show better tracking performance as well. For the 1000 Hz sinusoidal signal, the NRMSE for the 12-dimensional GP-based compensator is only 34.90% of that for the MPI compensator and 70.21% of that for the 2-dimensional GP-based compensator. For the 120 and 180 Hz mixed-frequency signal, the NRMSE for the 12-dimensional GP-based compensator is just 65.98% of that for the MPI compensator and 49.78% of that for the 2-dimensional GP-based compensator. The evaluation index values summarized in Table 2 strongly validate the effectiveness of the proposed high-dimensional GP-based compensator.

Comparisons of tracking accuracy for some representative signals with different compensators are shown in Figure 5. It can be observed that, even with complex signals, the proposed compensator still retains an excellent tracking accuracy as more information, including more temporal derivation as well as the temporal distribution, is introduced into the training sets compared to the normal 2-dimensional GP-based compensator and the inverse MPI based compensator. The superiority of this method is demonstrated clearly via comparing these tracking curves.

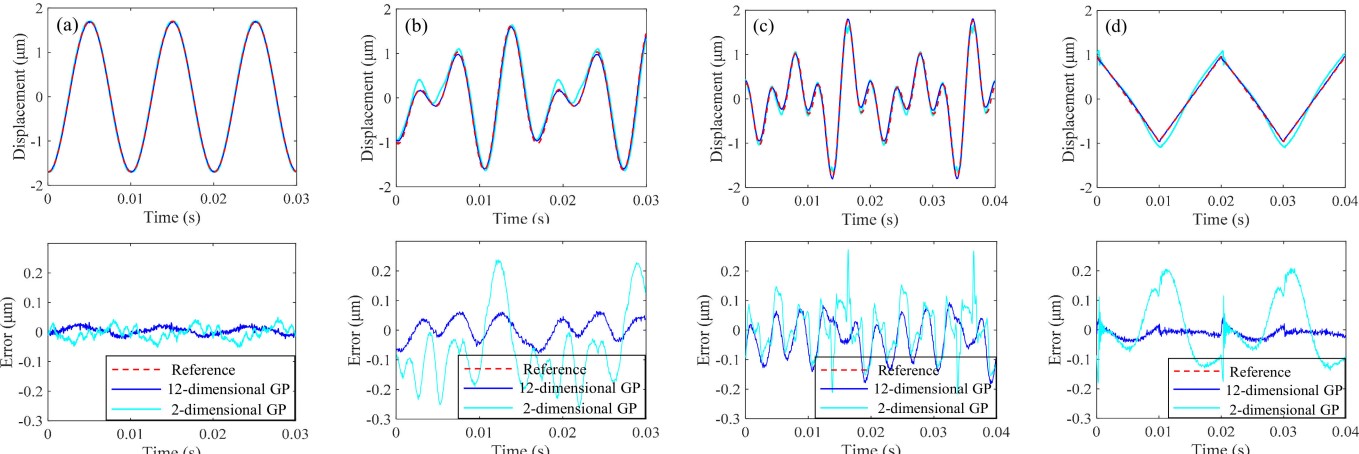

**Figure 5.** Tracking results of the open−loop controller under different reference signals: (**a**) 100 Hz sinusoidal signal; (**b**) dual−frequency mixed−signal; (**c**) four−frequency mixed−signal; (**d**) 50 Hz triangular wave.

### 3.2. Closed-Loop Controller

Although the GP-based IHC open-loop controllers perform well for the reference trajectories, especially when the dimensions of the input are high, it is difficult for them to cope with the creep of PEA and other kinds of external disturbances. Therefore, in order to improve the stability of the control system and the ability of the disturbance rejection, a feedforward/feedback closed-loop controller is designed based on the IHC, whose block diagram is shown in Figure 6. Here, $P(s)$ denotes the PEA system, and a feedback loop is used to eliminate the influence of the modeling error and various kinds of external disturbances. A widely used PI controller is utilized as the tracking controller $C(s)$. It can be written as:

$$v_{fb}(t) = K_p e(t) + K_i \int_0^t e(\tau)d\tau \tag{24}$$

where $e(t)$ is the tracking error, $K_p$ is the proportional gain and $K_i$ is the integral gain of the PI controller. It is known that the tracking performance of the PI controller becomes better with an increase in the control gain. However, overlarge control gains may cause a low relative stability margin. Hence, the specific parameters of $K_p$ and $K_i$ are finally maximized with the trial-and-error method in the step response experiments before the unstable vibration occurs.

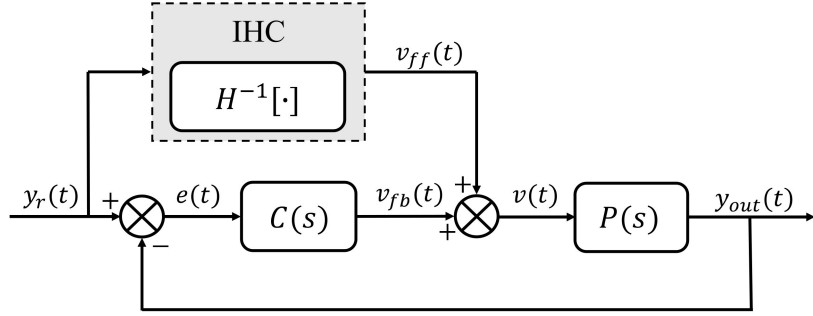

**Figure 6.** Block diagram of close-loop controller based on IHC.

Hence, the input voltage $v(t)$ of PEA is composed of two parts:

$$v(t) = v_{ff}(t) + v_{fb}(t) \tag{25}$$

where $v_{ff}(t)$ is the voltage generated from the IHC and $v_{fb}(t)$ is the voltage generated from the PI controller C(s).

To evaluate the performance of the feedforward/feedback closed-loop controller, the reference trajectories are selected the same as those used in the open-loop tracking tests. The pure PI controller without the feedforward compensator is also tested for comparisons. In the experiments, $K_p = 0.3$ and $K_i = 20,000$ for all controllers. The testing results are summarized in Table 3.

**Table 3.** Tracking results of the closed-loop controllers.

| The Type of Reference Trajectories | | Pure PI Controller (NRMSE/RME, %) | MPI-Based Method (NRMSE/RME, %) | GP-Based Method (NRMSE/RME, %) | | | |
|---|---|---|---|---|---|---|---|
| | | | | 2-Dimension | 6-Dimension | 9-Dimension | 12-Dimension |
| Sinusoid signal | 100 Hz | 1.1033/2.30 | 0.8094/2.15 | 0.7000/1.48 | 0.6940/1.62 | 0.6672/1.42 | 0.6407/1.38 |
| | 200 Hz | 2.2465/3.74 | 1.4492/2.73 | 1.4192/2.38 | 1.4005/2.43 | 1.3979/2.53 | 1.3883/2.48 |
| | 300 Hz | 3.1921/5.34 | 2.5918/6.57 | 3.0714/7.57 | 2.2458/4.17 | 2.1828/3.82 | 2.1721/3.86 |
| | 400 Hz | 4.2390/6.53 | 4.3373/12.4 | 4.2570/12.0 | 3.3804/5.70 | 3.2419/5.57 | 3.2052/5.44 |
| | 500 Hz | 5.3924/8.86 | 7.5462/17.6 | 5.1274/11.0 | 4.7590/8.32 | 4.2756/6.76 | 3.5501/6.86 |
| | 600 Hz | 6.5219/10.6 | 10.362/24.4 | 6.5440/12.4 | 5.8782/12.0 | 5.0000/9.19 | 4.9335/9.40 |
| | 700 Hz | 7.7353/12.3 | 10.020/23.3 | 7.2384/18.5 | 6.9444/14.1 | 6.6222/12.7 | 6.2461/10.8 |
| | 800 Hz | 8.9472/13.9 | 11.016/30.3 | 8.3979/19.4 | 7.5350/15.8 | 6.9765/12.7 | 6.6185/12.8 |
| | 900 Hz | 10.1254/16.2 | 11.556/32.6 | 8.8955/23.1 | 9.2630/19.1 | 7.7958/13.3 | 7.3941/11.9 |
| | 1000 Hz | 11.1463/18.2 | 14.633/31.5 | 9.4682/17.1 | 10.4491/20.4 | 9.05513/16.3 | 8.3210/14.6 |
| Mixed-frequency signal | 120 + 180 Hz | 1.3174/2.91 | 0.8590/2.09 | 0.8139/2.03 | 0.8012/2.05 | 0.7527/2.00 | 0.7497/1.84 |
| | 100 + 150 + 200 + 250 Hz | 2.2445/3.90 | 0.9088/2.87 | 1.1093/6.28 | 0.8894/2.64 | 0.8787/2.69 | 0.8065/2.43 |
| Triangular wave | 50 Hz | 1.3380/3.19 | 0.8782/2.84 | 0.9745/4.42 | 0.5786/3.22 | 0.4682/2.55 | 0.4405/2.41 |

From the table, it can be observed that the GP-based closed-loop controllers outperform the pure PI controller and the MPI-based closed-loop controller. The MPI-based controller performs better than the pure PI controller when tracking low-frequency sinusoidal trajectories, mixed-frequency trajectories and triangular trajectories since hysteresis is compensated to some extent. Nonetheless, when the tracking frequency becomes high, the tracking accuracy of this controller decreases rapidly and is even worse than that of the pure PI controller since it suffers from the rapid deterioration of its modeling accuracy. For the GP-based method, however, the better tracking accuracy can still be maintained at high frequencies due to its smaller modeling errors, especially for the models with 12-dimensional and 9-dimensional inputs. It is worth mentioning that for the mixed-frequency signal and triangular wave, with the increase in the input dimension of the model, the tracking errors decrease greatly. For the four-frequency mixed reference trajectory, the NRMSE for the 12-dimensional GP-based method is just 35.93% and 72.70% of those for the pure PI method and the 2-dimensional GP-based method, respectively. For the triangular reference trajectory, the NRMSE for the 12-dimensional GP-based method is just 32.92% and 50.16% of those for the pure PI method and 2-dimensional GP-based method, respectively. In order to better illustrate the performance of the proposed high-dimensional GP-based closed-loop controller, comparisons of tracking performance with different methods and input dimensions are plotted in Figure 7. From this figure, it can be observed that the results with the proposed controller outperform those obtained with the traditional 2-dimensional GP-based controller and the pure PI controller, which demonstrates its priority. In conclusion, the experimental results shown in the table and the figure validate the effectiveness of the high-dimensional GP-based method.

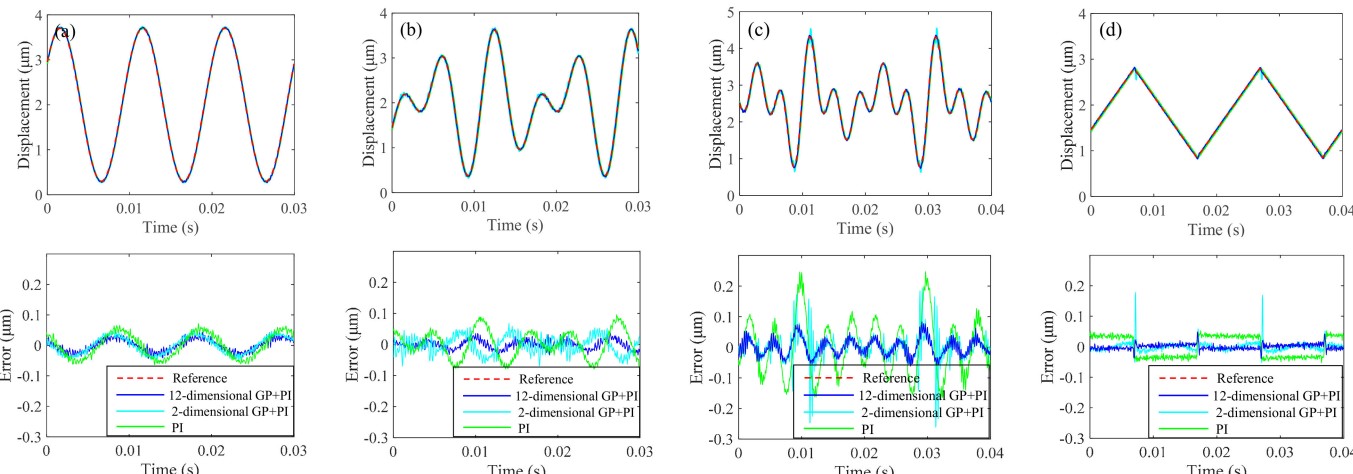

**Figure 7.** Tracking results of the closed−loop controller under different reference signals: (**a**) 100 Hz sinusoidal signal; (**b**) dual−frequency mixed−signal; (**c**) four−frequency mixed−signal; (**d**) 50 Hz triangular wave.

## 4. Conclusions

GP regression is a promising method to model rate-dependent hysteretic nonlinearity. However, when tracking high-frequency (i.e., higher than 500 Hz) or complex (i.e., mixed-frequency and triangular) signals, the accuracy of the traditional GP-based model with 2-dimensional input will deteriorate significantly. In this paper, more temporal derivative information, as well as the temporal distribution information, is introduced into the training set by increasing the input dimension of the GP model to improve its modeling accuracy. Experimental results show that with the increase in the input dimension, the prediction accuracy of the model improves greatly, especially for the complex signals, such as the mixed-frequency signal and the triangular wave. For the four-frequency mixed reference trajectory, the NRMSE for the 12-dimensional GP-based method is just 35.93% and 72.70% of those for the pure PI method and the 2-dimensional GP-based method, respectively. As these two types of trajectories are widely used in PEA applications, such as AFM and FTS, the improvement of modeling accuracy is of great importance. GP-based IHCs with different input dimensions are also constructed, and the open-loop and closed-loop controllers based on these IHCs are designed. Testing results show that the 12-dimensional GP-based open-loop and closed-loop controllers both exhibit the best performance among the similar controllers and the tracking performance is much better than that of the classical MPI based controllers, which validates the effectiveness of modeling the rate-dependent hysteresis of PEAs with high-dimensional input GPs.

**Author Contributions:** Y.M.: Methodology, Formal analysis, Investigation, Writing—Original Draft. X.W.: Formal analysis, Writing—Review and Editing. L.L.: Data curation. W.H.: Software. L.Z.: Conceptualization, Methodology, Project administration. All authors have read and agreed to the published version of the manuscript.

**Funding:** This paper was funded by the National Natural Science Foundation of China under Grant Nos. U2013211, 51975375 and 52105581, and the China Postdoctoral Science Foundation (No. 2021M692065).

**Institutional Review Board Statement:** Not applicable.

**Informed Consent Statement:** Not applicable.

**Data Availability Statement:** Not applicable.

**Acknowledgments:** This work was partially supported by the National Natural Science Foundation of China under Grant Nos. U2013211, 51975375 and 52105581, and the China Postdoctoral Science Foundation (No. 2021M692065).

**Conflicts of Interest:** The authors declare no conflict of interest.

**Appendix A. MPI Model for Rate-Dependent Hysteresis**

The play operator-based MPI model can be expressed as [27]:

$$y(t) = H[v](t) = g(v(t)) + \sum_{i=1}^{N} q_i F_{ori}^{h}[v](t) \qquad (A1)$$

where $y(t)$ denotes the output, $N$ is the number of the play operators, $F_{ori}^{h}(t)$ represents the play operator which has taken the rate term $\dot{v}(t)$ into account, $q_i$ denotes the corresponding weight of the $i$th play operator and $g(v(t)) = a_1 v^3(t) + a_2 v(t) + a_3$ is the modified term in order to describe the asymmetric hysteresis of the PEA with constant parameters $a_1$, $a_2$ and $a_3$.

In practical applications, $N$ is often set to be 10 and there are also two parameters in the play operator [27]. Therefore, altogether there are 15 parameters that need to be identified in the MPI rate-dependent model, which are obtained by particle swarm optimization. For more information about the MPI model, readers may refer to [27].

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
