# Peer review of "Hysteresis Modeling and Compensation of Piezoelectric Actuators Using Gaussian Process with High-Dimensional Input"

_actuators, doi:10.3390/act11050115_

Round 1

Reviewer 1 Report

The paper is devoted for hysteresis modeling and compensation of piezoelectric actuators using Gaussian process. The topic is generally interesting, however the paper contains unexplained places (below) and need major revisions.

Equations (1) and (2) should be more commented.

lines 270-271 "The specific parameters of Kp and Ki in the experiments are obtained by the trial and error method", please explain which criterion was used for  determination that values of these parameters are appropriate?

Figs. 1, 5 and 7 need more explanations.

In conclusions, line 310, "when tracking high-frequency" please explain meaning of "high-frequency".

All abbreviations should be explained by first using. For example, ADC, DAC.

English need minor revisions.

Reviewer 2 Report

The manuscript concerns the investigation of improvements in the models that should be used for PEAs and their rate-dependent hysteresis.

The innovative aspects of the manuscript should be better highlighted in the Introduction as well as in the Discussion. For example, highlight the difference with other papers on the same topic that proposed also other approaches (e.g., doi: 10.1016/j.heliyon.2020.e03999; doi: 10.1109/AMC.2014.6823364; Kui Li et al 2019 Smart Mater. Struct. 28 115038).

References should be updated.

Detailed comments are reported here below.

- line 30: better explain if the value here reported (35%) is considered nonadequate for the accuracy of the positioning and which value could be considered as the threshold

- lines 33-34: add at least two references for “researchers ….”

- lines 55-57: add some references related to machine learning for this particular application

- lines 81-85: the paragraph does not add any relevant information to the manuscript and should be removed

- line 151: specify the material of the PEA

- lines 174-176: give a reference for the choice of these evaluation indexes

- lines 186-…: try to deeply discuss the obtained results and differences between the two models, the strong points of the approach you are here proposing

- Table 2 and related text: also, in that case, try to evidence the difference between the inverse MPI model and 2D GP-based inverse hysteresis model, deeper describing the reasons for that

- lines 289-…: see the previous comments

Reviewer 3 Report

The paper on Hysteresis modeling and compensation of piezoelectric actuators using Gaussian process with high-dimensional input by Y. Meng et al. 

presents an inverse hysteresis compensator implemented  both in open- and closed-loop controllers for a hysteresis control of piezoelectric actuators. The results show that increasing to higher dimension inputs can definitely help to increase position accuracy.The paper is correctly written and presented with quality format and plots. The content is relevant to show the details of the methodology, the discussion is sound and the results are clear. The interest of the paper is clear. However, before publication, some points must be addressed to make a better version, with updated information.

Review comments and Review points are provided to the Authors for revision.

- - - - - REVIEW COMMENT - - - - - - 

-1-

The main concern is about the connection of sections 2.4 and 3. The discussion and training of the model in Section-2 was based in voltage time-distribution and the measured displacement. In Section-3, the Authors refer to measured displacement  time-distribution and to voltage. Is this right? Am I missing something? 

If so, it is not stated clearly the connection between sections and the model used in the two sections. Maybe the Authors wanted to show the sequence of 1/ making the model based in voltage time-distribution, and 2/ using the same model and just 'inverting' the model terms to input the displacement time-dependences to define the voltage. If so, this game should be explained much more clearly than just eq-22. 

Explain clearly, with some detail in Section-3 and present to Readers the whole methodology. Also connect the information on the hyper-parameter set (eq-11) obtained in Section-2 and the model applied in Section-3.

-2-

Note that the PEA travel range is about 26 um ( L151), however the results presented are in a range below 9 um. The nonlinearity effects are well-known to depend on amplitude, as the Authors point in L29.  Therefore, it is important to declare why the range 0-30 V (thus,  0-9 um) was selected in this study. At least, simply state that this study was dedicated to the 0-10 um range as a reference for any range study. 

Reviewer 4 Report

The authors present a modified GP regression to model the rate-dependent hysteretic nonlinearity. Temporal derivative information as well as the temporal distribution information is introduced into the training set by increasing the input dimension of the GP model to improve its modeling accuracy. Experimental results show that with the increase of the input dimension, the prediction accuracy of the model improves greatly, especially for complex signals such as the mixed-frequency signal and the triangular wave. The topic matter is quite interesting and technologically important. The manuscript is well-structured and well-written. However, it can be further improved by properly addressing the below comments:

1) The main novelty of the present work should be highlighted. Please also clearly state the literature gap of the current study vis-à-vis the existing literature in the introduction clearly.

2) The authors should mention the effect of the neural network parameters (such as learning rate, and activation function) on the performance of the developed approach. Parametric studies may be conducted to gain a deep insight into the importance of these parameters on the accuracy of the neural network outputs.

3)  The authors should emphasize the reasons for choosing the recurrent neural network, instead of the more general-purpose fully-connected neural network approach. The following papers can be commented on in the introduction to make the reference more complete.

Raj, R.A., Samikannu, R., Yahya, A. and Mosalaosi, M., 2020. Performance evaluation of natural esters and dielectric correlation assessment using artificial neural network (ANN). Journal of Advanced Dielectrics, 10(05), p.2050025.

Haggag, S., Nasrat, L. and Ismail, H., 2021. ANN approaches to determine the dielectric strength improvement of MgO based low density polyethylene nanocomposite. Journal of Advanced Dielectrics11(4), pp.2150016-48727.

Round 2

Reviewer 1 Report

Authors make proper corrections according to reviewer remarks

and I suggest publish the paper as it is.

Author Response

Thanks again for your positive comments and kind suggestions.

Reviewer 3 Report

The Reviewer thanks the interest of the Authors for providing an improved version, including information and details that are important for better discussing and understanding the paper.

Few details can still improve the paper in some parts. Please, consider.

I believe that some information on the details provided in the point-10- are worth to mention to have the full picture of the methodology used in the paper. It could be kind of: (just before the sentence on L199 The results obtained with...)

The hyperparameters set was obtained by a typical GP algorithm implemented in our MATLAB environment, converged after some 200 iterations on the measured data. No specific parameter optimization was applied at this stage of the investigation, focused in the adequate performance of the GP method.

Figure 3 

It is unclear that (a)-(b)-(c)-(d) refers to the 4 columns. Therefore, consider to remark that in the caption, as:

Experimental measured results of the PEA (red lines) as function of voltage and time for  500 Hz sinusoidal signal (column a panels) ; 1000 Hz sinusoidal signal (column b panels) ; four-frequency mixed-signal  (column c panels); 50 Hz triangular wave  (column d panels). The prediction results for 2-dimensional and 12-dimensional GP model are also plot (blue lines) in the upper two and lower two panels, respectively.

Consider to use a bigger size and bold-format for  (a)-(b)-(c)-(d) labels.

L150

It is worth noting that the original signal collected by the displacement sensor exists noise.

Check the meaning:

It is worth noting that the original signal collected by the displacement sensor includes noise.
